# Antimicrobial effects of inhaled sphingosine against *Pseudomonas aeruginosa* in isolated ventilated and perfused pig lungs

**Henning Carstens**[1,2]*, **Katharina Kalka**[1], **Rabea Verhaegh**[3], **Fabian Schumacher**[4], **Matthias Soddemann**[3], **Barbara Wilker**[3], **Simone Keitsch**[3], **Carolin Sehl**[3], **Burkhard Kleuser**[4], **Michael Hübler**[2], **Ursula Rauen**[5], **Anne Katrin Becker**[3], **Achim Koch**[1], **Erich Gulbins**[3,6]☯, **Markus Kamler**[1]☯

1 Department of Thoracic and Cardiovascular Surgery, University Hospital Essen, University of Duisburg-Essen, Essen, Germany, 2 Cardiac Surgery for Congenital Heart Disease, University Medical Center Hamburg-Eppendorf, Hamburg, Germany, 3 Institute of Molecular Biology, University of Duisburg-Essen, Essen, Germany, 4 Institute of Pharmacy, Freie Universität Berlin, Berlin, Germany, 5 Institute of Biochemistry, University of Duisburg-Essen, Essen, Germany, 6 Department of Surgery, University of Cincinnati, Medical School, Cincinnati, OH, United States of America

☯ These authors contributed equally to this work.
* h.carstens@uke.de

**Data Availability Statement:** All relevant data are within the paper and its Supporting Information files.

## Abstract

### Background

*Ex-vivo* lung perfusion (EVLP) is a save way to verify performance of donor lungs prior to implantation. A major problem of lung transplantation is a donor-to-recipient-transmission of bacterial cultures. Thus, a broadspectrum anti-infective treatment with sphingosine in EVLP might be a novel way to prevent such infections. Sphingosine inhalation might provide a reliable anti-infective treatment option in EVLP. Here, antimicrobial potency of inhalative sphingosine in an infection EVLP model was tested.

### Methods

A 3-hour EVLP run using pig lungs was performed. Bacterial infection was initiated 1-hour before sphingosine inhalation. Biopsies were obtained 60 and 120 min after infection with *Pseudomonas aeruginosa*. Aliquots of broncho-alveolar lavage (BAL) before and after inhalation of sphingosine were plated and counted, tissue samples were fixed in paraformaldehyde, embedded in paraffin and sectioned. Immunostainings were performed.

### Results

Sphingosine inhalation in the setting of EVLP rapidly resulted in a 6-fold decrease of *P. aeruginosa* CFU in the lung (p = 0.016). We did not observe any negative side effects of sphingosine.

**Funding:** The study was supported by DFG grant GU 335/39-1 to EG and KA 5448/1-1 to MK. The funders had no role in study design, data collection and analysis, decision to publish, or preparation of the manuscript.

**Competing interests:** The authors have declared that no competing interests exist.

**Abbreviations:** AP, alkaline phosphatase; AT1, alveolar type 1; AT2, alveolar type 2; BAL, broncho alveolar lavage; BEC, bronchial epithelial cells; Cer, ceramide; CF, cystsic fibrosis; CFU, colony forming unit; EVLP, ex-vivo lung perfusion; FCS, fetal calf serum; FiO$_2$, inspired oxygen; LDH, lactate dehydrogenase; LTx, lung transplantation; MDR, multi drug resistance; MS, mass spectrometry; PA, Pseudomonas aeruginosa; PAP, pulmonary artery pressure; PEEP, positive endexpiratory pressure; PVR, pulmonary vascular resistance; SA, Staphylokokkus aureus; SM, sphingomyelin; SPH, sphingosine.

## Conclusion

Inhalation of sphingosine induced a significant decrease of *Pseudomonas aeruginosa* at the epithelial layer of tracheal and bronchial cells. The inhalation has no local side effects in *ex-vivo* perfused and ventilated pig lungs.

## Introduction

Lung transplantation (LTx) is still the gold standard in the treatment of patients suffering from end-stage lung diseases with more than 4500 annual registered LTx worldwide [1]. Despite an improvement in survival over time, the current median survival is only approximately 6 years and therefore worse than the expected survival in other solid organ transplantations [1]. According to the International Heart and Lung Transplantation Registry (ISHLT), infections are the second leading cause of mortality within the first 30 days (17.2%) rising to the main cause of death within the first year after LTx (33.1%) [2]. In this context, it is important to note that in contrast to other solid organ transplants with superior outcomes, the lungs are continuously exposed to pathogens. Most important pathogens in post-transplant patients are *Pseudomonas aeruginosa* (PA), *Staphylococcus aureus* (SA), Enterobacteriaceae, Klebsiella spp., *Escherichia coli*, Acinetobacter, *Enterococcus faecalis*, *Candida albicans*, Aspergillus spp., cytomegalovirus and herpes simplex virus [3–6]. PA is also known as a common pathogen among patients suffering from chronic obstructive pulmonary disease, bronchiectasis or cystic fibrosis (CF) [7, 8]. In the context of hospital acquired or ventilator associated pneumonia PA is the leading pathogen for severe courses [9]. Over the last decade an increase of multidrug-resistance (MDR) in bacterial pathogens like PA, *Acinetobacter species*, *Staphylococcus aureus* (SA) and others was recorded [10, 11]. This "antibiotic crisis" as the WHO headlined [12] needs to be addressed and research into new antibacterial approaches is necessary. Sphingosine (SPH) has been recently identified as a lipid with marked antimicrobial potency. Sphingosine is markedly reduced in the respiratory tract of CF mice and patients and reconstitution of sphingosine in lungs of CF mice or in CF humans restores bacterial defense in these mice [13–17]. This sphingoid long-chain base is thus part of the innate immune system. It is synthesized from ceramide by the enzyme ceramidase [from a review see e.g. 18]. The antimicrobial potency has been proven by several *in vitro* and *in vivo* studies in which sphingosine showed high efficiency against several bacterial species including PA, SA (even methicillin resistant SA; MRSA), *Acinetobacter baumannii*, *Escherichia coli*, and *Neisseria meningitides* [13–16, 19–23]. In previous studies we showed that inhalation of sphingosine led to a dose dependent uptake of sphingosine into bronchial epithelial cells (BEC) in mini pigs. This increase of sphingosine levels in the luminal membrane of BEC and trachea was without a concomitant systemic accumulation or any side effects up to very high doses [24, 25].

Due to the fact that donor lungs used for LTx are frequently positive for bacterial cultures and bacterial populations (46–89%), which possibly leads to a donor-to-recipient transmission with a subsequently higher risk of lung infection and therefore a reduced posttransplant outcome [26–28], we here aimed to analyze whether inhalation of sphingosine into *ex-vivo* perfused and ventilated domestic pig lungs (EVLP) reduces or even eliminates lung infection with *Pseudomonas aeruginosa*. In addition, EVLP systems like XVIVO® built an optimal platform where marginal donor-lungs can easily be subjected and be treated with antibiotics in high dose regimes [29, 30] or maybe other anti-infective agents like sphingosine prior to lung re-implantation.

## Methods

### Animals

Mature domestic male hybrid pigs (BW 30–35 kg) were used with supervision of the central animal laboratory of the University of Duisburg-Essen. Human care was provided in compliance to the "Principles of Laboratory Animal Care' formulated by the National Society for Medical Research and the 'Guide for the Care and Use of Laboratory Animals' prepared by the Institute of Laboratory Animal Resources and published by the National Institutes of Health (NIH Publication No. 86–23, revised 1996). Prior to euthanasia no medication was administered.

### Ethics statement

Experiments were designed as organ procurements, which were reported to local authorities (Landesamt für Natur, Umwelt und Verbraucherschutz NRW) according to applicable law (§ 1 VTMVO). We confirm that all experiments were performed in accordance with the relevant guidelines (including the ARRIVE guidelines) and regulations.

### Lung procurement and EVLP settings

Anesthesia of male domestic hybrid pigs (bodyweight 35+/- 5kg) induced with ketamine (30 mg/kg BW i.m.) (Ursotamin®, Serumwerk Bernburg AG, Bernburg, Germany) and xylazin (2 mg/kg BW i.m.) (Xylavet®, cp-pharma®, Burgdorf, Germany) sedation followed by intravenous application of midazolam (0.5 mg/kg BW i.v.) (midazolam-ratiopharm®, ratiopharm® GmbH, Ulm, Germany) and again ketamine (30 mg/kg BW i.v.) after insertion of an i.v. line. This was followed by final euthanasia using a potassium chloride overdose (7.45%, 1.7 ml/kg BW i.v.) (Kaliumchlorid 7.45%, B. Braun Deutschland GmbH & Co.KG, Melsungen, Germany). A midline sternotomy was used for a standard lung procurement as described elsewhere [31, 32]. Thereafter lungs were then immediately stored cold for one hour. An acellular solution (MP-Custodiol-MP, Dr. Franz Köhler Chemie GmbH, Bensheim, Germany) was used for the EVLP circuit. Custodiol-MP base solution was supplemented with the lyophilisate supplied by the manufacturer, 300 ml of 20% low-sodium human serum albumin (HSA 20%, CSL Behring GmbH, Marburg, Germany), 3000 IU heparin (heparin-natrium 25000ratiopharm, ratiopharm GmbH,Ulm,Germany) and 35 ml of 5% glucose (G5, B. Braun, Melsungen, Germany) yielding final concentrations of 23 μM deferoxamine, 6.8 μM LK614, 8.8 mM glucose and 54.5 g/l albumin. Lungs were randomized in two groups, i.e. one group receiving saline 0.9%: (n = 4; one additional pig was excluded due to positive testing for typical porcine pathogens prior to any experiment); and one group receiving 500 μM sphingosine (C18-sphingosine, Avanti Polar Lipids, Inc., AL, USA) (n = 5). A modified Toronto protocol with a pressure-controlled ventilation mode was used to ensure a gentle initial recruitment of atelectatic lungs [33, 34]. Both lung-groups were ventilated according to pressure controlled ventilation during EVLP (Inspiratory"peak" pressure of 16 mbar; I:E 4:1, 6 cm $H_2O$ PEEP and RR of 6 breaths/min) with an endotracheal tube of 8.0 mm. For the 1st recruitment maneuver (1 h after infection) and 2nd recruitment maneuver (1 h after inhalation) ventilation settings were increased (PEEP 8 cm $H_2O$, inspiratory pressure 20 cm $H_2O$, RR 8 breaths/min) for a period of 10 minutes. For measurements during recruitments fraction of inspired oxygen (FiO$_2$) was increased from 21% up to 100%. All lungs remained on EVLP for 3 hours. For statistical analysis mean values during the 10-minute recruitment maneuvers were compared between both groups. At the end of EVLP, a dorsal biopsy of the right lung was taken for pathology and for calculation of wet-to-dry weight (W/D) ratio and total lung weight was measured after

disconnection from the circuit. Analysis of the perfusate was performed during the above-named points to measure lactate dehydrogenase (LDH) and alkaline phosphatase (AP) activity.

### Infection and inhalation

*Pseudomonas aeruginosa* strain ATCC 27853 was used for the pneumonic infection. Growth of bacterial cultures was conducted as previously described by us [15]. Bacteria were diluted into pre-warmed saline 0.9% solution to a concentration of $2 \times 10^9$ CFU per 5 mL. This dose was shown to induce a severe infection in EVLP lungs in our pretests. To ensure an equal distribution of bacterial strains within the lungs, a nebulizer (Aerogen Ultra, Aerogen Limited, Galway, Ireland) connected to the endotracheal tube was used (Fig 1A). Sphingosine inhalation via an ultrasound nebulizer (Aeroneb®, Aerogen Limited, Galway, Ireland) was started 1 h after infection. Lungs were treated by a 15-minute nebulization of a 5 mL sphingosine suspension containing 500 μM sphingosine in 0.9% saline or a 5 ml 0.9% saline as control (Fig 1B).

### Biopsy and Broncho-Alveolar Lavage (BAL)

A fiberoptic videoscope (Ambu A/S, Baltorpbakken 13, DK-2750 Ballerup, Denmark) was used to extract BAL and biopsies (Fig 2A and 2B). BAL samples were taken by applying 10 mL of saline 0.9%. To avoid wash-out phenomena different target areas for the BAL were used. Out of each BAL an amount of 100 μL was pipetted onto tryptic soy agar plates and bacterial colonies were counted after incubation for 18 h at 37°C. Biopsies were processed as previously reported [25].

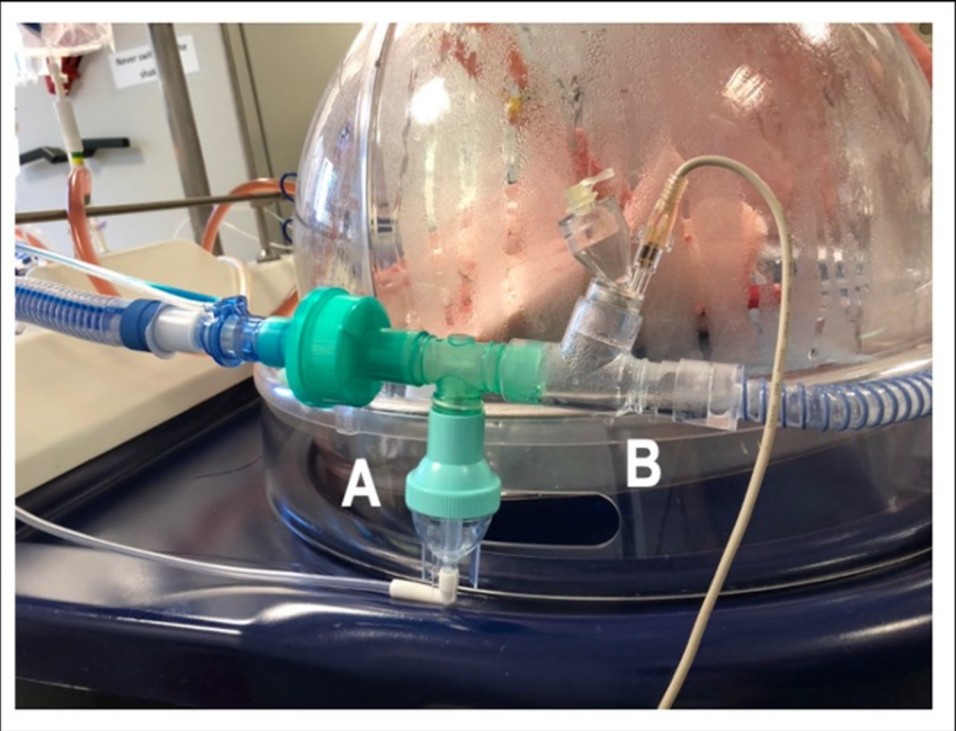

**Fig 1.** Nebulizer for the bacterial strains (A) and ultrasound nebulizer for the inhalation solution (B).

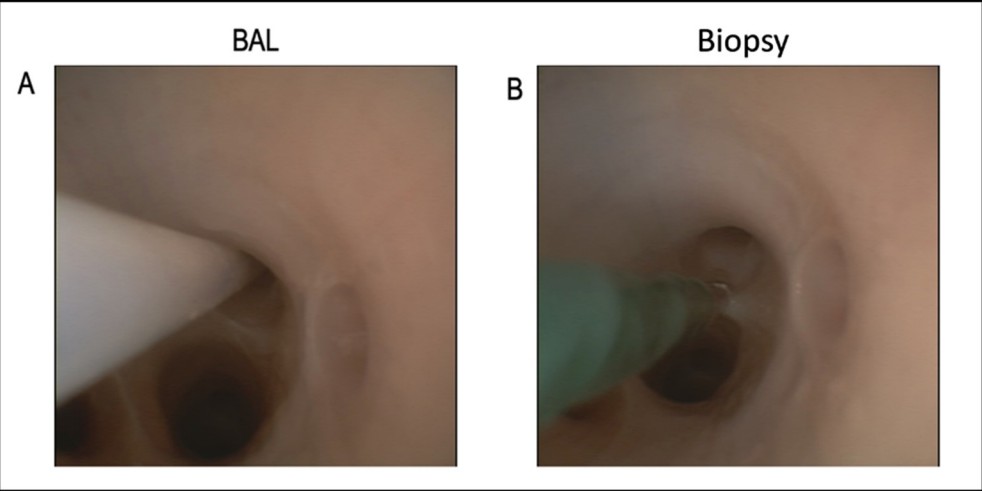

**Fig 2.** A and B: Target areas for broncho-alveolar lavage (A) and biopsies (B).

## Quantification of sphingosine in BAL

BAL (200 µL) were extracted in $CHCl_3/CH_3OH/1N$ HCl (100:200:1, v/v/v). The lower phase was dried and resuspended in a detergent solution consisting of 7.5% (w/v) n-octyl glucopyranoside, 5 mM cardiolipin in 1 mM diethylenetriamine-pentaacetic acid. The kinase reaction was started by addition of 0.001 units sphingosine kinase (R&D) in 50 mM HEPES (pH 7.4), 250 mM NaCl, 30 mM $MgCl_2$ 10 µM ATP and 10 µCi $[^{32}P]\gamma$ATP. Samples were incubated for 30 min at 37˚C with shaking (350 rpm) followed by extraction in 20 µL 1N HCl, 800 µL $CHCl_3:CH_3OH:1N$ HCl (100:200:1, v/v/v), and 240 µL each of $CHCl_3$ and 2 M KCl. Phases were separated, the lower phase was collected, dried, dissolved in 20 µL $CHCl_3:CH_3OH$ (1:1, v/v), and separated on Silica G60 TLC plates with $CHCl_3:CH_3OH$:acetic acid:$H_2O$ (90:90:15:5, v/v/v/v). The TLC plates were analyzed with a phosphorimager. Sphingosine levels were determined with a standard curve of C18-sphingosine.

## Quantification of sphingosine, ceramide and sphingomyelin by HPLC-MS/MS

Tissue specimens were subjected to lipid extraction as performed previously in our group [35]. The internal standards $d_7$-sphingosine ($d_7$-SPH), 17:0 ceramide and 16:0-$d_{31}$-sphingomyelin (all Avanti Polar Lipids, Alabaster, USA) were added to the extraction solvent. Lipids were chromatographically separated. A 6490 triple-quadrupole mass spectrometer (Agilent Technologies, Waldbronn, Germany) was used for MS/MS detection after positive electrospray ionization (ESI+) of sphingolipid analytes. SPH and six sub-species (16:0, 18:0, 20:0, 22:0, 24:0 and 24:1) each of ceramides (Cer) and sphingomyelins (SM) were analyzed [36]. Sphingolipid quantities were normalized to the protein content of the tissue homogenates used for extraction, as determined via Bradford assay, and expressed as "pmol/mg protein".

## Immunohistochemistry

Tissue samples from EVLP pig lungs were stained for sphingosine, ceramide or neutrophils/monocytes using the anti-GR1 antibodies as previously described [15–17]. Immediately after removal, the tissue specimens were fixed in 4% paraformaldehyde (PFA), subsequently dehydrated with an ethanol-to-xylol gradient, embedded in paraffin, sectioned at 7 µm, dewaxed

and rehydrated. The sections were then washed, blocked for 10 min at room temperature with 5% fetal calf serum (FCS) and incubated with anti-sphingosine (1:1000 dilution) (clone NHSPH, #ALF-274042010, Alfresa Pharma Corporation), anti-ceramide (1:100 dilution) (clone S58-9, #MAB_0011, Glycobiotech) or anti-Ly-6G/Ly-6C (anti-GR1) (1:200) (clone RB6-8C5; #553122; BD Pharmingen) antibodies in HEPES buffered saline (H/S, 132 mM NaCl, 20 mM HEPES [pH 7.4], 5 mM KCl, 1 mM $CaCl_2$, 0.7 mM $MgCl_2$, 0.8 mM $MgSO_4$) plus 1% FCS. Afterwards the samples were washed again and then with Cy3-coupled anti-mouse IgM F(ab)$_2$ fragments (Jackson Immunoresearch) or Cy3-coupled anti-rat IgG F(ab)$_2$ fragments (Jackson Immunoresearch). Finally, samples were embedded in Mowiol. Granulocytes in the epithelial cell layer were quantified. Evaluation of the stainings was performed using confocal microscopy (Leica TCS-SP5 confocal microscope). Images were analyzed with the Leica LAS AF software (Leica Microsystems, Mannheim, Germany). Control stainings were performed with irrelevant mouse IgM or rat IgG followed by the corresponding Cy3-coupled secondary antibody. Fluorescence intensities were quantified using Image J.

### Hemalaun stainings

Lung paraffin tissue samples from EVLP pigs were dewaxed, rehydrated and washed as above. Sections were stained with hemalaun for 5 min. Afterwards, the samples were embedded in Mowiol and analyzed using a Leica TCS-SP5 confocal microscope equipped with a 40x lens. A classifying score was used for analysis.

### Statistics

Comparisons were made between inhalation groups receiving sphingosine 500 μM (SPH 500) or saline 0.9% (NaCl 0.9%). Data was explored in mean value (mean) and standard deviation (sd). Differences were considered significant at the level of $p < 0.05 = *$, $p < 0.01 = **$, students t-test was used for normal distribution otherwise Mann-Whitney-u-test was applied. Statistical analysis was performed using SPSS Statistics 22 (IBM, Armonk, New York, US).

### Results

To determine the number of *P. aeruginosa* in the lung after infection, EVLP lungs of pigs were infected and we obtained BAL samples 1 h after the infection (Figs 1 and 2A). We analyzed aliquots of 0.1 mL of these BAL samples by culturing them on agar plates and report the CFU of these cultures as a measurement for bacterial numbers in the lung. Analysis of the CFU on agar plates yielded counts of 665 ± 538 CFU in the sphingosine group and 531 ± 393 CFU in the saline group 1 h after infection, verifying a consistent infection in both groups (Fig 3). Streak cultures from BAL fluid obtained one hour after inhalation of sphingosine contained a significantly (p = 0.016) reduced number of *P. aeruginosa* in the lung with 119 ± 138 CFU (Fig 3). Please note that in 2 out of 5 pigs sphingosine reduced the number of bacteria in the lung under the detection level. In contrast, inhalation of 0.9% NaCl did not reduce numbers of *P. aeruginosa* in the lung 1 h after the inhalation, which even slightly increased from 531 ± 393 CFU to 618 ± 397 CFU (Fig 3).

To validate the increase of sphingosine levels in the bronchial system of the lungs after inhalation, sphingosine amounts were measured in the BAL samples. After inhalation of NaCl 0.9% solution no increase of sphingosine was observed in the sodium group between "after infection" and "after inhalation" (9.0±1.7 vs. 8.6±0.8; p = 0.7), whereas the samples in the sphingosine group showed a significant increase after inhalation of 500 μM sphingosine solution from "after infection" to "after inhalation" (8.4±2.4 vs. 18.7±6.1; p = 0.008) as shown in Fig 4.

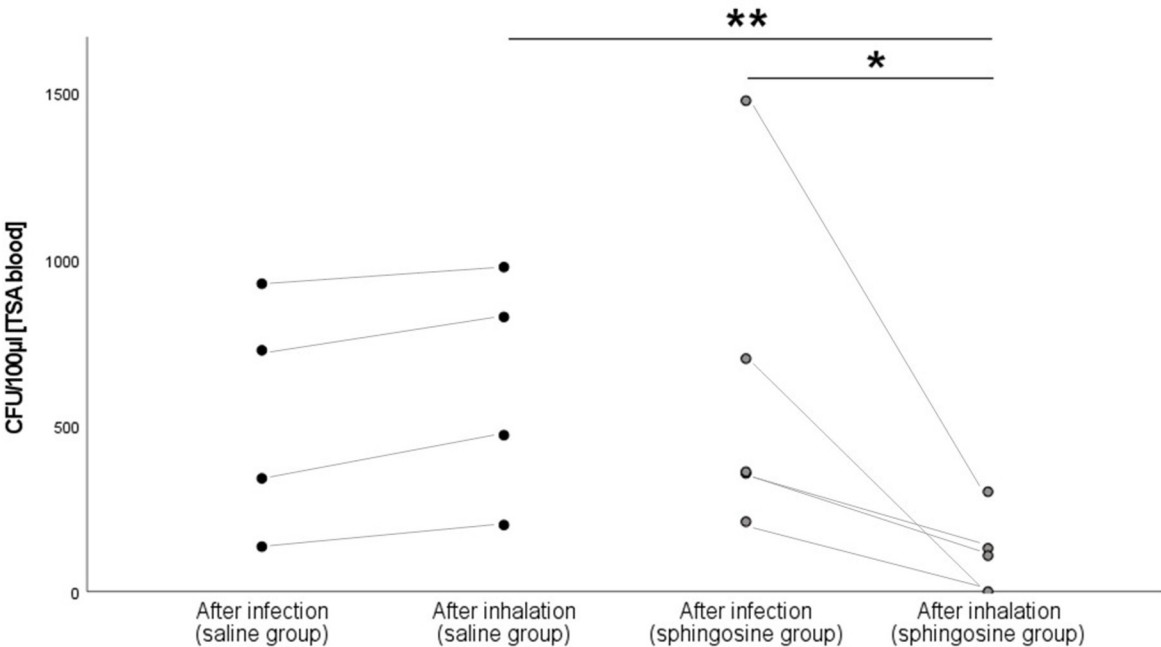

**Fig 3. Analysis of CFU of streak cultures from BAL fluid (100 μL/plate).** Pigs were infected with $2 \times 10^9$ CFU of *P. aeruginosa* (ATCC 27853), BAL was performed with 10 mL (NaCl 0.9%), 5 mL were aspirated, collected and 0.1 mL aliquots were plated on tryptic soy agar plates with 5% sheep blood (Groups: "After infection"). CFU were counted after an incubation period of 18 h at 37˚C. Sphingosine inhalation (500 μM in 5 mL 0.9% NaCl) significantly reduced CFU of *P. aeruginosa*, while inhalation of 0.9% NaCl was without effect (Groups: "After inhalation"). Given are dot-plots with connecting lines, n = 4 for the 0.9% control group and n = 5 for the sphingosine group: $^*p < 0.05$, $^{**}p < 0.01$; students t-test/ Mann-Whitney-u-test.

Mass spectrometry (MS) analysis of bronchial biopsies revealed a slight, but not significant increase of sphingosine, ceramide and sphingomyelin concentrations after inhalation of 500 μM SPH (Table 1). Samples from different lung compartments did not reveal a statistically significant difference in measured concentrations between the groups receiving 0.9% NaCl or 500 μM sphingosine. The variation of extracted samples was rather high, very likely because these biopsies contained very variable amounts of epithelial cell layer vs. submucosa and MS determines sphingosine not only in the epithelial cells that are exposed to inhaled sphingosine, but also endogenous sphingosine in the submucosa and other bronchial tissues (Table 1).

To allow analysis of sphingosine and ceramide specifically in bronchial epithelial cells, we performed immuno-histological studies on our tissue samples. To this end, we stained paraffin sections with Cy3-coupled monoclonal anti-sphingosine antibodies. The analysis revealed a significant increase in fluorescence intensity of sphingosine in bronchial epithelial cells after tube-inhalation of a 500 μM SPH suspension compared to saline (NaCl 0.9%) solution (Fig 5).

Next, a possible conversion from sphingosine into ceramide within bronchial epithelial cells was studied. Sections were stained with Cy3-coupled monoclonal anti-ceramide antibodies and analyzed by confocal microscopy. The studies did not show a significant change of ceramide concentrations in the sphingosine group in comparison to the saline group (Fig 6).

Next, pro-inflammatory effects were studied. To this end, sections were stained with Cy3-coupled anti-Gr1-antibody. Analysis did not reveal a significant difference in influx of granulocytes/monocytes after tube-inhalation of 500 μM sphingosine compared to the saline group (Fig 7A). In order to investigate whether sphingosine affected epithelial cell integrity in infected lungs, sections of epithelial cell layers were stained with hemalaun. The results

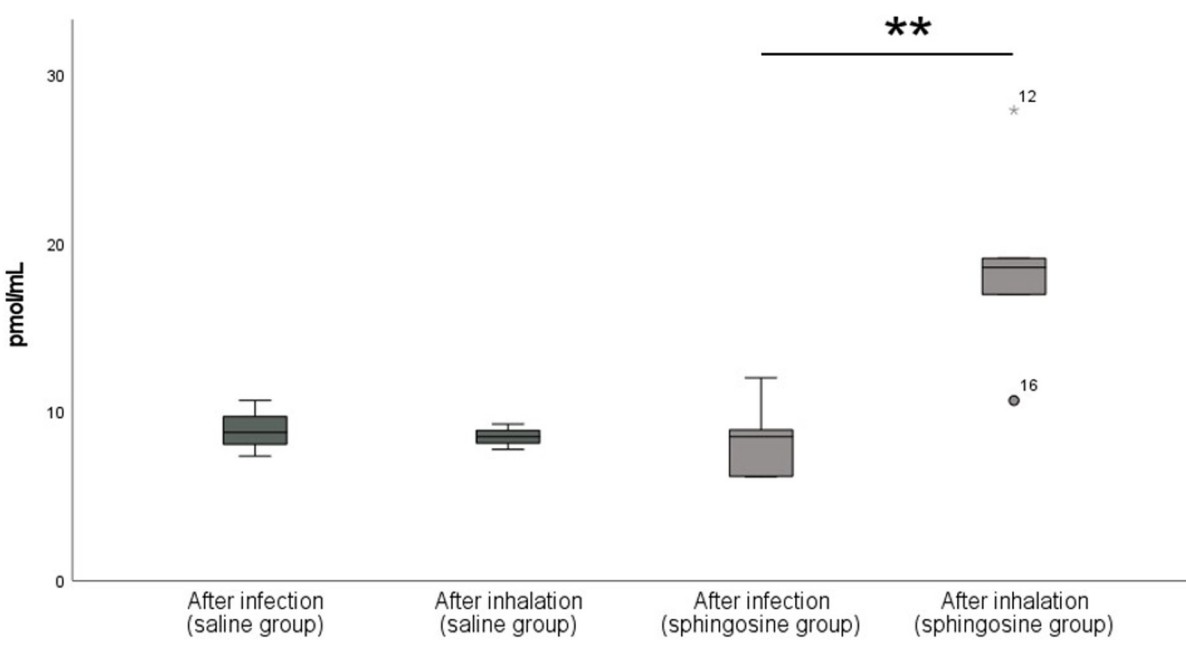

**Fig 4. Analysis of sphingosine levels from BAL.** After a 15-minute nebulization of a 5 mL sphingosine suspension containing 500 μM sphingosine in 0.9% saline or 5 ml 0.9% saline as control a fiberoptic videoscope was used to extract BAL. BAL samples were taken by applying 10 mL of saline 0.9% to the bronchial system. Analysis of sphingosine levels in the sodium group resulted in comparable sphingosine levels "after infection" and "after inhalation" (9.0±1.7 vs. 8.6±0.8; p = 0.7). Whereas sphingosine levels significantly increase in the sphingosine group from "after infection" to "after inhalation" (8.4±2.4 vs. 18.7±6.1; p = 0.008). Given are dot-plots with connecting lines, n = 4 for the 0.9% control group and n = 5 for the sphingosine group: *p < 0.05, **p < 0.01; students t-test/ Mann-Whitney-u-test.

demonstrate that tube-inhalation with sphingosine did not affect integrity of bronchial epithelial cells (Fig 7B).

Analysis of functional lung parameters during EVLP revealed an impaired lung function after application of $2x10^9$ CFU of *Pseudomonas aeruginosa* ATCC 27853, with an increased lung resistance and decreasing values of oxygenation capacity and lung compliance. Oxygenation capacity ($PO_2/FiO_2$ ratio), compliance (Cstat, Cdyn), pulmonary vascular resistance (PVR) and pulmonary artery pressure (PAP) did not significantly differ between lungs that were inhaled with sphingosine or 0.9% NaCl as control (Table 2).

No relevant group differences in wet/dry ratio and lung weights before and after EVLP run were detected (Table 3).

Perfusate measurements revealed an increasing activity of alkaline phosphatase (AP) and lactate dehydrogenase (LDH) from first to second measurements without statistically significant differences between both groups (Table 4).

## Discussion

The present study indicates that sphingosine is a safe and potent antibacterial treatment option for pulmonary *P. aeruginosa* infections and potentially also for other bacterial pneumonia. We demonstrate that a 15-minute inhalation of 500 μM sphingosine has the capability to reduce CFU of *P. aeruginosa* by a factor of almost 6. No significant side effects in EVLP pig lungs were observed, in accordance with previous *in-vivo* inhalation studies in infected and non-infected mice and non-infected mini pigs [15, 22, 24, 37]. As reported in our previous study, a 14-day period of sphingosine inhalation twice daily in a non-infected mini-pig model studying potential adverse effects of sphingosine did not result in obvious changes of health status nor did it

**Table 1. Mass spectrometry analysis of total sphingosine, ceramide and sphingomyelin in biopsies and tissue samples from EVLP pig lungs after infection and inhalation of sphingosine (500 μM SPH) or saline (NaCl 0.9%).**

| | NaCl 0.9% (n = 4) | 500 μM SPH (n = 5) | |
| --- | --- | --- | --- |
| | mean±SD | mean±SD | p-value |
| *1st biopsy* | | | |
| Sphingosine [pmol/mg protein] | 48±3 | 53±14 | n.s. |
| Ceramide total [pmol/mg protein] | 1212±632 | 1429±310 | n.s. |
| Sphingomyelin total [pmol/mg protein] | 11960±5830 | 17052±3766 | n.s. |
| *2nd biopsy* | | | |
| Sphingosine [pmol/mg protein] | 44±1 | 83±56 | n.s. |
| Ceramide total [pmol/mg protein] | 1300±440 | 2137±1243 | n.s. |
| Sphingomyelin total [pmol/mg protein] | 18207±3409 | 31863±20570 | n.s. |
| *Peripheral bronchial tissue* | | | |
| Sphingosine [pmol/mg protein] | 197±72 | 190±55 | n.s. |
| Ceramide total [pmol/mg protein] | 2051±834 | 1520±452 | n.s. |
| Sphingomyelin total [pmol/mg protein] | 45715±4934 | 41072±10698 | n.s. |
| *Distal bronchus tissue* | | | |
| Sphingosine [pmol/mg protein] | 43±12 | 50±33 | n.s. |
| Ceramide total [pmol/mg protein] | 1325±413 | 1426±681 | n.s. |
| Sphingomyelin total [pmol/mg protein] | 15589±2512 | 16227±5265 | n.s. |
| *Main bronchus tissue* | | | |
| Sphingosine [pmol/mg protein] | 52±35 | 49±45 | n.s. |
| Ceramide total [pmol/mg protein] | 1524±533 | 1288±499 | n.s. |
| Sphingomyelin total [pmol/mg protein] | 12075±399 | 12083±3000 | n.s. |

Levels of sphingosine, ceramide and sphingomyelin were determined in bronchoscopy biopsies 1 h after infection and 1 h after inhalation. Analyzed were each 1 sample/pig with 5 pigs in the sphingosine group and 4 pigs in the saline group. Given is the mean ± SD of each parameter, p<0.05 was considered as significant, not significant was abbreviated with n.s.; students t-test/ Mann-Whitney-u-test.

lead to any changes in the lung or local signs of inflammation in the airway tract [24]. The present experimental setup allowed the application of a well-defined and evenly distributed number of bacterial pathogens and amount of sphingosine, respectively, into the broncho-alveolar system. In addition to this aspect of direct tube inhalation, the setup offers the opportunity to measure lung function continuously. However, despite the reduction of CFU after sphingosine inhalation, functional parameters did not differ between infected groups that were inhaled with 0.9% saline or 500 μM sphingosine. We assume that the time course of only 1 h observation of the lungs after inhalation is too short to restore the functions of endothelial cells and changes in the lung parenchyma once disturbed after infection. For instance, it is known that lung alveolar type I cells and alveolar type II cells regulate the structural integrity and function of alveoli after *P. aeruginosa* induced acute lung injury and restoration of alveolar integrity may require more time. Furthermore, an extensive release of lipopolysaccharides due to the sphingosine induced cell damage of *Pseudomonas aeruginosa* is possible and might have some effects on functional lung parameters with a delayed restoration of these functions.

Sphingosine kills pathogens within a few minutes by inducing membrane permeabilization [38] and thus, even a short period of 1 h observation after infection is more than sufficient to allow killing of the pathogens.

The measurements of sphingosine in the BAL suggest that most of the inhaled sphingosine remains in the fluid on top of the epithelial cells. In accordance, we were unable to measure a

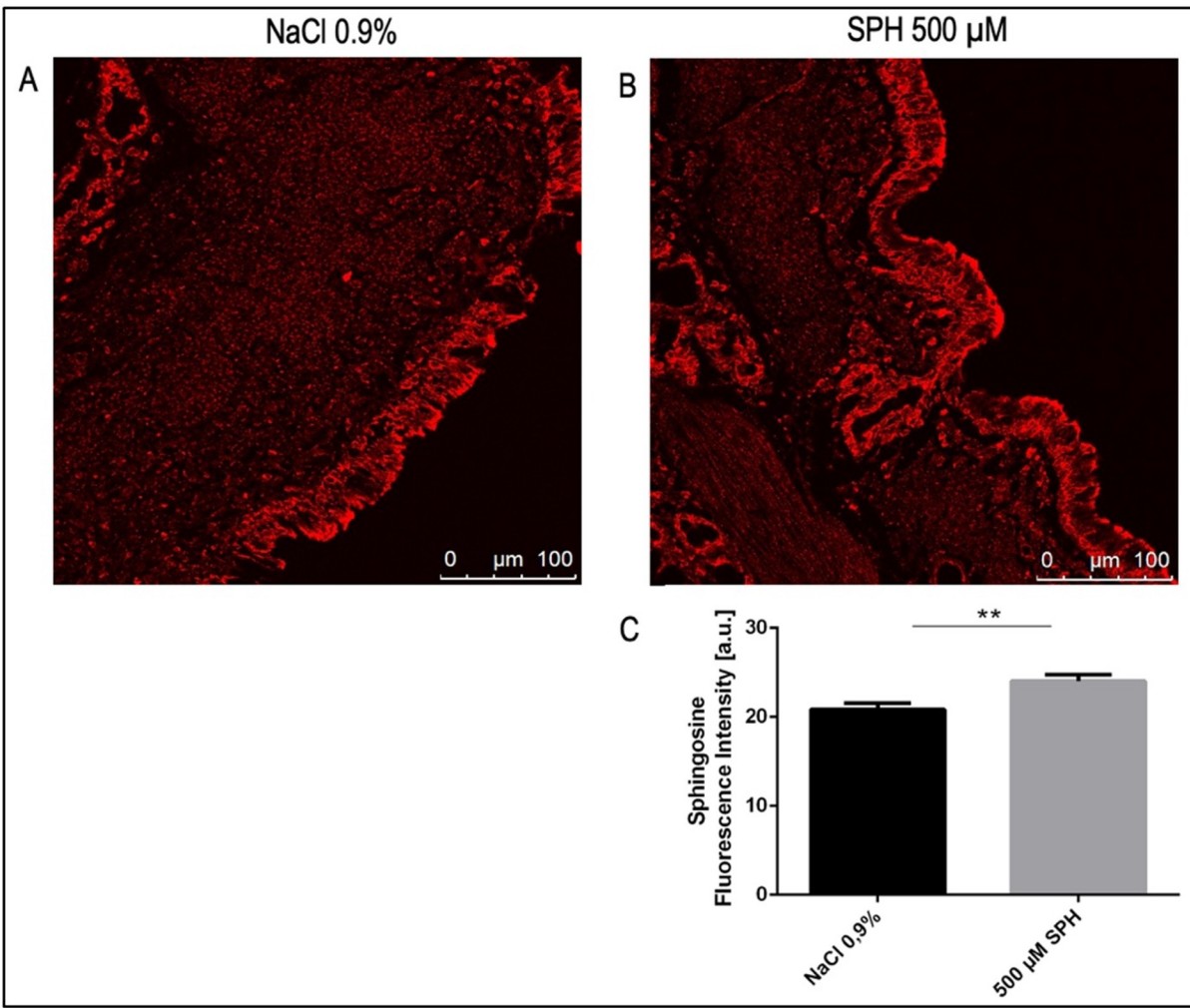

**Fig 5. Histological studies of lungs stained with Cy3-coupled anti-sphingosine antibodies.** Sphingosine concentrations increased in bronchial epithelial cells after inhalation of a 500 µM solution. EVLP pig lungs were inhaled with sphingosine (n = 5) or saline (0.9% NaCl) as control group (n = 4). Biopsies were fixed in paraformaldehyde, embedded in paraffin and sectioned. Sections were stained with Cy3-coupled anti-sphingosine antibodies. Shown are representative immune stainings for the saline group (A) and the sphingosine group (B) and quantitative analysis of the fluorescence intensity of a total of (C). Given is the mean ± SD of sphingosine fluorescence intensity from each 3 sections with analysis of fluorescence intensity in epithelial cells within 5 visual fields per animal (n = 4 for the control group and n = 5 for the sphingosine group), **p<0.01; students t-test/ Mann-Whitney-u-test.

significant increase of sphingosine in biopsies and only a small increase of sphingosine in fluorescence microscopy stainings. These stainings allowed to determine the concentration of lipids within cells, which is important to determine side effects of the inhalation. The higher concentration of sphingosine in the BAL indicates that inhaled sphingosine is able to reach in particular extracellular bacteria in the lung.

Confocal microscopy was performed to measure sphingosine and ceramide concentrations in bronchial epithelial cells after inhalation of a 500 µM SPH containing solution compared to saline. Our studies demonstrate a significantly increased sphingosine concentration in epithelial cells of infected EVLP pig lungs after administration of sphingosine, while ceramide levels did not change. However, sphingosine levels in epithelial cells only moderately increased and given the strong effect of sphingosine on *P. aeruginosa*, we assume that most of the inhaled sphingosine remained in the mucus and the bronchial epithelial fluid lining the epithelial cells

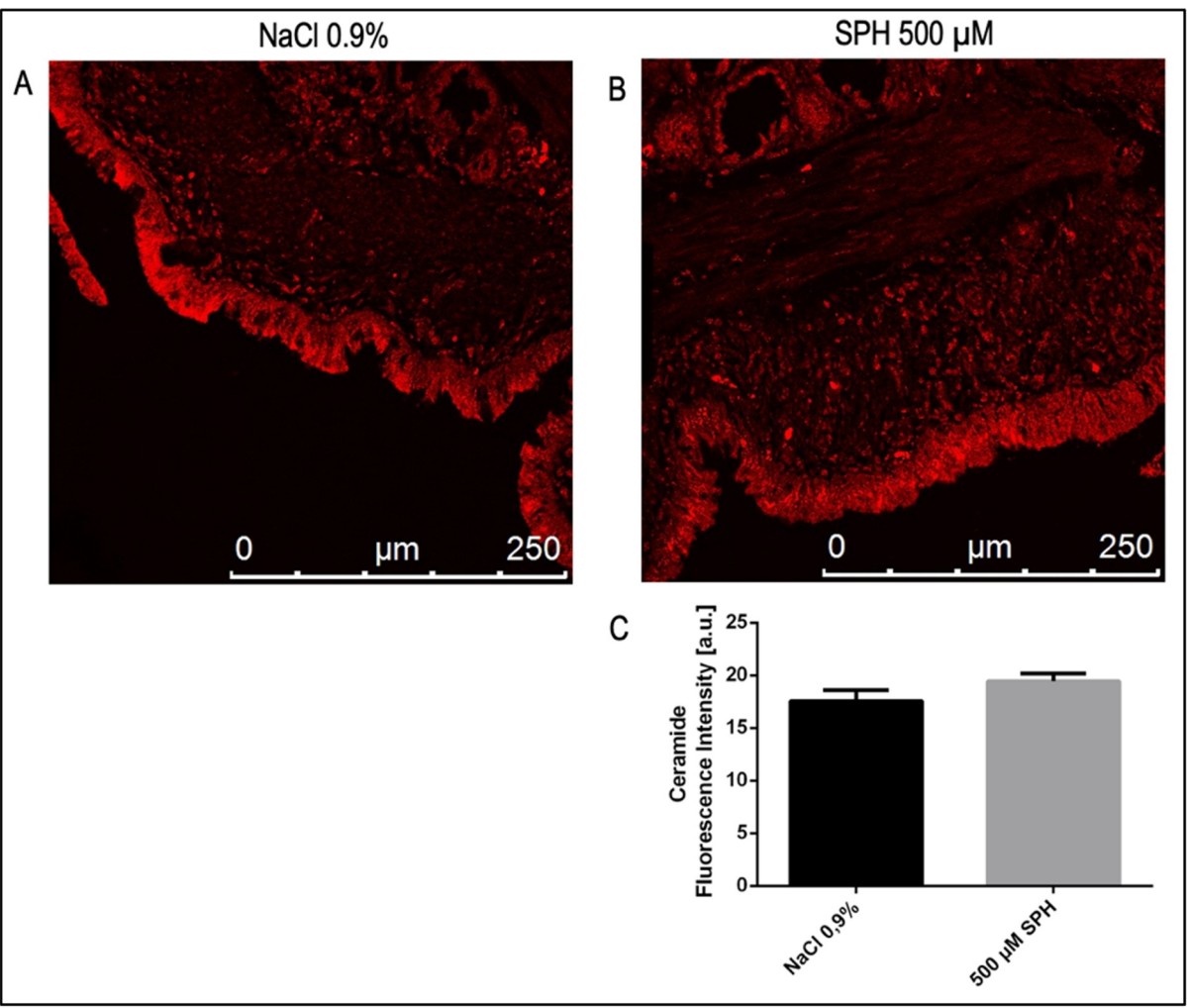

**Fig 6. Histological studies of lungs stained with Cy3-coupled anti-ceramide antibodies.** No differences in ceramide of BEC were detected after sphingosine inhalation (n = 5 pigs) compared to saline (NaCl 0.9%) inhalation (n = 4 pigs) (A, B). Quantitative analysis of the fluorescence intensity of a total of 1200 cells per group (C). Given is the mean ± SD of ceramide fluorescence intensity from each 3 sections with analysis of fluorescence intensity in epithelial cells within 5 visual fields per animal (n = 4 for the control group and n = 5 for the sphingosine group), p<0.05 was considered as significant; students t-test/ Mann-Whitney-u-test.

as discussed above. This fluid is lost during fixation of the lung tissues, but previous studies have shown that most of inhaled sphingosine remains in this fluid and the mucus [39]. Thus, sphingosine will be very active against extracellular bacteria such as *P. aeruginosa*, while most of the sphingosine does not seem to reach the epithelial cells, increasing the safety of sphingosine inhalations. The long chain base sphingosine is part of the innate defense against pathogens and is integrated in the lipid composition of the bronchial epithelial cell layer [15, 17, 39]. Previous studies demonstrated that low levels of sphingosine are sufficient to treat or prevent pulmonary infections, at least in mice, in particular mice with cystic fibrosis [15, 17, 39]. Cystic fibrosis mice exhibit a constitutive decrease of sphingosine in bronchial epithelial cells, which is caused by an IRF-8-regulated down-regulation of acid ceramidase expression in Cftr-deficient bronchial epithelial cells [40]. The lack of sphingosine resulted in increased infection susceptibility of CF-mice, which was corrected by inhalation of sphingosine [40].

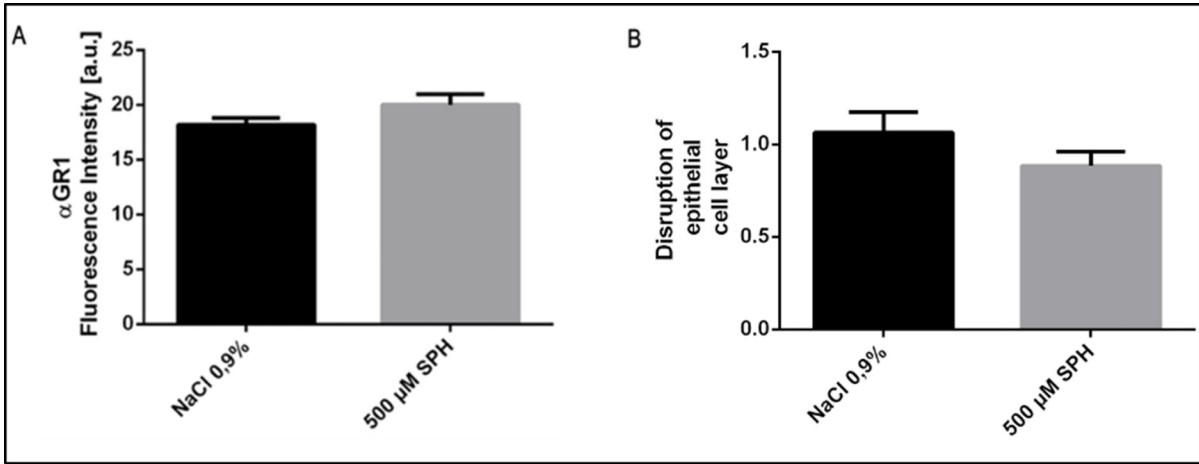

**Fig 7.** A and B: Quantitative analysis of histological studies from EVLP lungs stained with Cy3-coupled anti-Gr1-antibodies (αGR1). (A) No significant difference in granulocytes and monocytes influx into the epithelial cell layer was observed. Quantitative analysis of the fluorescence intensity of a total of 1200 bronchial epithelial cells per group is displayed. Quantitative analysis of the disruption of epithelial cell layer (B) after inhalation of NaCl 0.9% or 500 µM SPH. No significant differences in epithelial cell integrity using a grading model (Grade 0: no change of the epithelial cell layer, basal membrane intact, less than 2% pyknotic, i.e. dead epithelial cells. Grade 1: small disruptions of the epithelial cell layer, basal membrane intact, less than 5% pyknotic, i.e. dead epithelial cells; Grade 2: Larger disruptions of the epithelial cell layer, basal membrane still intact, less than 10% pyknotic, i.e. dead epithelial cells; Grade 3: Larger disruptions of the epithelial cell layer, disrupted basal membrane, more than 10% pyknotic, i.e. dead epithelial cells) were observed between the groups. Given is the mean ± SD of αGR1 Fluorescence intensity and disruption of epithelial cell layer from each 3 sections with analysis of fluorescence intensity in epithelial cells within 5 visual fields per animal (n = 4 for the control group and n = 5 for the sphingosine group), p<0.05 was considered as significant; students t-test/ Mann-Whitney-u-test.

**Table 2. Analysis of functional lung parameters during EVLP run.**

| | | NaCl 0.9% (n = 4) | SPH 500 µM (n = 5) | |
| --- | --- | --- | --- | --- |
| | | Mean ± SD | Mean ± SD | p-values |
| $PO_2/FiO_2$ ratio [mmHg] | After infection | 174±72 | 168±68 | n.s. |
| | After inhalation | 133±37 | 114±65 | n.s. |
| | p-values | n.s. | n.s. | |
| Dynamic compliance | After infection | 12±7 | 8±7 | n.s. |
| [mL mbar$^{-1}$] | After inhalation | 9+4 | 7±2 | n.s. |
| | p-values | n.s. | n.s. | |
| Static compliance | After infection | 16±13 | 15±16 | n.s. |
| [mL mbar$^{-1}$] | After inhalation | 9±4 | 9±5 | n.s. |
| | p-values | n.s. | n.s. | |
| PVR [mmHg L$^{-1}$min m$^2$] | After infection | 1968±561 | 1830±355 | n.s. |
| | After inhalation | 2073±973 | 2098±464 | n.s. |
| | p-values | n.s. | n.s. | |
| PAP [mmHg] | After infection | 25±7 | 24±2 | n.s. |
| | After inhalation | 25±6 | 25±1 | n.s. |
| | p-values | n.s. | n.s. | |

Values 1-h after infection with *Pseudomonas aeruginosa* strain (ATCC 27853) and 1 h after inhalation of NaCl 0.9% (n = 4) or SPH 500 µM (n = 5) reveals no statistically significant differences between both groups. Parameters containing oxygenation capacity (PO$_2$/FiO$_2$ ratio), dynamic and static lung compliance, pulmonary vascular resistance (PVR) and pulmonary artery pressure (PAP). Given are the means ± SD, p<0.05 was considered as significant, not significant was abbreviated with n.s.; students t-test/ Mann-Whitney-u-test.

**Table 3. Analysis of wet/dry ratio and total weight gain during a 3 h EVLP run.**

| | NaCl 0.9% (n = 4) | 500 µM SPH (n = 5) | |
|---|---|---|---|
| | mean±SD | mean±SD | p-values |
| W/D ratio | 8.9±2.2 | 8.2±1.9 | n.s. |
| Lung weight before EVLP | 427±42 | 407±45 | n.s. |
| Lung weight after EVLP | 705±199 | 602±147 | n.s. |

No significant differences were detected between both groups. Given is the mean ± SD, p<0.05 was considered as significant, not significant was abbreviated with n.s.; students t-test/ Mann-Whitney-u-test.

**Table 4. Analysis of perfusate measurements during EVLP run.**

| | NaCl 0.9% (n = 4) | 500 µM SPH (n = 5) | |
|---|---|---|---|
| | mean±SD | mean±SD | p-value |
| Alkaline phosphatase (AP) | | | |
| After infection | 2.3±1.5 | 3.2±2.4 | n.s. |
| After inhalation | 5.8±6.3 | 5.8±5.9 | n.s. |
| Lactate dehydrogenase (LDH) | | | |
| After infection | 147±88 | 178±30 | n.s. |
| After inhalation | 258±128 | 218±65 | n.s. |

Values 1 h after infection with *Pseudomonas aeruginosa* (ATCC 27853) and 1 h after inhalation of NaCl 0.9% (n = 4) or SPH 500 µM (n = 5) showed no statistically significant differences between both groups. Given is the mean ± SD, p<0.05 was considered as significant, not significant was abbreviated with n.s.; students t-test/ Mann-Whitney-u-test.

The present study supports the notion that sphingosine is also an effective new therapeutic treatment option against bacterial colonization or bacterial infection of EVLP lungs prior to transplantation. No side effects were observed and a broad-spectrum antibacterial activity was already published in small animal experiments [14, 15, 17, 22, 24]. Further research is required to investigate the antibacterial effects to other pathogens in EVLP lungs.

## Conclusion

In summary, we demonstrate that infection of an EVLP pig lung with $2x10^9$ CFU of *P. aeruginosa* strain ATCC 27853 leads to a pronounced acute pneumonia with decreased lung function. Furthermore, a 15-minute treatment of these lungs with sphingosine via tube-inhalation decrease CFU of *P. aeruginosa* by a factor of almost 6. Further studies on several other P. aeruginosa strains are necessary to generalize our findings.

## Supporting information

**S1 Checklist. The ARRIVE guidelines 2.0: Author checklist.**
(PDF)

## Author Contributions

**Conceptualization:** Henning Carstens, Erich Gulbins.

**Data curation:** Henning Carstens, Katharina Kalka.

**Formal analysis:** Henning Carstens, Katharina Kalka, Rabea Verhaegh.

**Funding acquisition:** Erich Gulbins, Markus Kamler.

**Investigation:** Henning Carstens.

**Methodology:** Henning Carstens, Fabian Schumacher, Matthias Soddemann, Barbara Wilker, Simone Keitsch, Carolin Sehl, Burkhard Kleuser.

**Project administration:** Henning Carstens, Katharina Kalka, Erich Gulbins, Markus Kamler.

**Resources:** Henning Carstens, Katharina Kalka, Rabea Verhaegh, Ursula Rauen.

**Software:** Henning Carstens.

**Supervision:** Erich Gulbins.

**Writing – original draft:** Henning Carstens.

**Writing – review & editing:** Katharina Kalka, Burkhard Kleuser, Michael Hübler, Ursula Rauen, Anne Katrin Becker, Achim Koch, Erich Gulbins, Markus Kamler.

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
