## [Decision Letter · Decision Letter 0]

4 Apr 2022

PONE-D-22-06001Antimicrobial effects of inhaled sphingosine against Pseudomonas aeruginosa in isolated ventilated and perfused pig lungsPLOS ONE

Dear Dr. Henning Carstens,

Thank you for submitting your manuscript to PLOS ONE. After careful consideration, we feel that it has merit but does not fully meet PLOS ONE’s publication criteria as it currently stands. Therefore, we invite you to submit a revised version of the manuscript that addresses the points raised during the review process.

We look forward to receiving your revised manuscript.

Kind regards,

Abdelwahab Omri, Pharm B, Ph.D, Full Professor, Laurentian University

Academic Editor

PLOS ONE

Journal Requirements:

2. As part of your revision, please complete and submit a copy of the Full ARRIVE 2.0 Guidelines checklist, a document that aims to improve experimental reporting and reproducibility of animal studies for purposes of post-publication data analysis and reproducibility: https://arriveguidelines.org/sites/arrive/files/Author%20Checklist%20-%20Full.pdf (PDF). Please include your completed checklist as a Supporting Information file. Note that if your paper is accepted for publication, this checklist will be published as part of your article.

Reviewers' comments:

Reviewer's Responses to Questions

**Comments to the Author**

1. Is the manuscript technically sound, and do the data support the conclusions?

Reviewer #1: Yes

Reviewer #2: Partly

2. Has the statistical analysis been performed appropriately and rigorously? 

Reviewer #1: Yes

Reviewer #2: Yes

3. Have the authors made all data underlying the findings in their manuscript fully available?

Reviewer #1: Yes

Reviewer #2: Yes

4. Is the manuscript presented in an intelligible fashion and written in standard English?

Reviewer #1: Yes

Reviewer #2: Yes

5. Review Comments to the Author

Reviewer #1: Carstens et al. demonstrated antimicrobial activities of sphingosine against Pseudomonas aeruginosa infection during EVLP. The manuscript is well written and makes a clear point. But I do have some concerns that need the authors to help clarify.

1. Line 195: please avoid starting a sentence with a number such as 100uL.

2. In all tables, please clearly indicate n.s.=non-significant.

3. The authors used a typical P. aeruginosa (ATCC 27853) reference strain for ex vivo tests, which is fine for proof of concept studies. But most nosocomial P. aeruginosa infections are likely multi-drug resistant. Have the authors considered using P. aeruginosa clinical isolates that are known to be antibiotic-resistant and see if sphingosine would show similar antimicrobial activities? At least the authors could put in more details on this topic in their discussion and cite previous research.

4. It’s a bit strange that the authors decided to go directly with ex vivo studies without a preliminary in vitro test of sphingosine activity against P. aeruginosa. For any potential therapeutic option, we need to at least see their MIC. Is 500uM sphingosine much higher or lower than its MIC against P.a. ATCC27853?

5. To claim sphingosine as a potential drug candidate for P. aeruginosa reduction during EVLP, a screening of sphingosine MIC against multiple clinical P. aeruginosa isolates is necessary. Or the authors should point out that it is a limitation of the study by only testing P.a. ATCC27853 ex vivo.

6. Fig. 3: are there any cases where the CFU decreased to 0 after 15 minutes of sphingosine treatment? Without a dot plot, it’s hard to interpret the data. Complete CFU elimination, even in rare cases, is encouraging because untreated P. aeruginosa almost always leads to chronic lung infections. Given the current data, sphingosine is not significantly disrupting lung functions even at 500uM. Would an increase of sphingosine dosage help achieve P. aeruginosa elimination? Please at least add some discussions.

7. Table 4: it’s quite difficult to distinguish the difference between NaCl and SPH treated groups by looking at the tissue images. The difference in fluorescence intensity, although showed statistical significance, seems small. The authors mentioned that the treatment time might not be long enough for SPH to enter epithelial cells, which could be the case. But I presume the BAL was collected before the lungs are fixed in paraformaldehyde? Have the authors considered using ELISA to quantify the exact concentrations of SPH/ceramide in BAL? Before the lungs are fixed, the authors could also have homogenized some fresh tissue samples and performed a side-by-side SPH/ceramide comparison between BAL and tissue.

Reviewer #2: Henning Carstens and co-authors submitted an article on the evaluation of nebulized sphingosine to treat a pig model of perfused lung infected with Pseudomonas aeruginosa. The main result of the study was the possibility of decreasing the bacterial burden in perfused pig lungs by a factor of 6 without observing any side effects.

The study was well conducted and several experiments were performed to evaluate the effect of nebulized sphingosine.

Main points

My main point is that in general, reductions in bacterial load are expressed in logs, and a 1-log (10-fold) reduction is somewhat considered small. Perhaps this point should be explained in the manuscript?

The second point is that the authors showed that sphingosine is more effective than saline in reducing bacterial load, but no comparison was made with an antibiotic commonly used to treat Pseudomonas lung infections, such as ciprofloxacin or nebulized tobramycin.

Minor points

Why did you choose a concentration of 500 µM of sphingosine, what was the dose delivered to the lungs?

Line 186. I am not sure if the lungs can be inhaled. Perhaps write that the lungs were treated by nebulization of 5 ml of ...

Sphingosine was delivered as a suspension, did you evaluated the efficacy of the nebulization process (total output, particle size change during nebulization…).

In Fig. 3, why did you make 2 groups of untreated infected lungs. If all groups received the same, homogeneous initial bacterial inoculum, the CFU counts of the so-called "after infection" groups could be averaged to form a pre-treatment group to compare with the 2 treated groups (saline and sphingosine). Again, I think that a group treated with an antibiotic such as tobramycin could have helped evaluate the efficacy of the sphingosine treatment.

Why sphingosine was assayed only in tissues and not in the BAL? How long after the nebulization was collected the samples for sphingosine assays.

Why were the variabilities in sphingosine, ceramide, and sphingomyelin concentrations higher in the sphingosine group than in the saline group?

Lung functional parameters could have been tested on non-infected lungs before and after SPH nebulization to assess its effect on the lungs.

It could have been interesting to test several sphingosine doses and different Pseudomonas aeruginosa strains.

Line 418-419 is not clear to me.

6. PLOS authors have the option to publish the peer review history of their article (what does this mean?). If published, this will include your full peer review and any attached files.

Reviewer #1: No

Reviewer #2: **Yes: **frederic Tewes

---

## [Author Response · Author response to Decision Letter 0]

12 Jun 2022

Reviewer #1: Carstens et al. demonstrated antimicrobial activities of sphingosine against Pseudomonas aeruginosa infection during EVLP. The manuscript is well written and makes a clear point. But I do have some concerns that need the authors to help clarify.

1. Line 195: please avoid starting a sentence with a number such as 100uL.

- The text has been changed to “Out of each BAL an amount of 100 µL…”

2. In all tables, please clearly indicate n.s.=non-significant.

- Table legends were supplemented by “not significant was abbreviated with n.s.”

3. The authors used a typical P. aeruginosa (ATCC 27853) reference strain for ex vivo tests, which is fine for proof of concept studies. But most nosocomial P. aeruginosa infections are likely multi-drug resistant. Have the authors considered using P. aeruginosa clinical isolates that are known to be antibiotic-resistant and see if sphingosine would show similar antimicrobial activities? At least the authors could put in more details on this topic in their discussion and cite previous research.

- We agree that many of the nosocomial Pseudomonas strains are resistant, but there are also many Pseudomonas infections with a minor change in antibiotic resistance levels in the clinical setting. We initially chose a sensitive Pseudomonas strain to limit variables in an overall complex trial design, but plan to conduct further series of trials to test the efficacy of SPH in this setting against other bacterial strains and Pseudomonas strains with altered resistance status. However, we have previously shown in vitro and in mouse studies that sphingosine also kills P. aeruginosa and S. aureus that are resistant to antibiotics (Verhaegh R, Becker KA, Edwards MJ, Gulbins E. Sphingosine kills bacteria by binding to cardiolipin. J Biol Chem 2020;295:7686-96. Seitz AP, Schumacher F, Baker J, Soddemann M, Wilker B, Caldwell CC, Gobble RM, Kamler M, Becker KA, Beck S, Kleuser B, Edwards MJ, Gulbins E. Sphingosine-coating of plastic surfaces prevents ventilator-associated pneumonia. J Mol Med (Berl) 2019;97:1195-1211. Pewzner-Jung Y, Tavakoli Tabazavareh S, Grassmé H, Becker KA, Japtok L, Steinmann J, Joseph T, Lang S, Tuemmler B, Schuchman EH, Lentsch AB, Kleuser B, Edwards MJ, Futerman AH, Gulbins E. Sphingoid long chain bases prevent lung infection by Pseudomonas aeruginosa. EMBO Mol Med 2014;6:1205-1.)

4. It’s a bit strange that the authors decided to go directly with ex vivo studies without a preliminary in vitro test of sphingosine activity against P. aeruginosa. For any potential therapeutic option, we need to at least see their MIC. Is 500uM sphingosine much higher or lower than its MIC against P.a. ATCC27853?

- We have previously demonstrated the inhalative effect of SPH against P. aeruginosa in a several mouse model, including pulmonary infection of cystic fibrosis mice with several P. aeruginosa strains (and S. aureus) as well as in mouse models of pneumonia after burn or trauma. In these mouse models we demonstrated that inhalation of sphingosine at a concentration of 125 �M was sufficient to kill P. aeruginosa in murine lungs, even in chronically infected lungs. 

- We have also shown that sphingosine kills many bacterial strains in vitro. In these studies we demonstrated that sphingosine inhibited growth of and/or killed P. aeruginosa with an EC50 of 0.5 ± 0.1 mM, Acinetobacter baumannii with an EC50 of 0.07 ± 0.05 mM, of Moraxella catarrhalis of 0.04 ± 0.004 mM of Haemophilus influenzae with an EC50 of 4.8 ± 0.49 mM and Burkholderia cepacia with an EC50 of 45 ± 6.3 mM (Pewzner-Jung et al, 2014). This 500 mM is much higher than the MIC or EC50 concentration, but this is the concentration in the inhalation fluid and not the concentration reached in the bronchoalveloar fluid or on the cells. 

5. To claim sphingosine as a potential drug candidate for P. aeruginosa reduction during EVLP, a screening of sphingosine MIC against multiple clinical P. aeruginosa isolates is necessary. Or the authors should point out that it is a limitation of the study by only testing P.a. ATCC27853 ex vivo.

- We now discuss this issue mentioning the references above and we also state that a limitation of the present work is that the demonstrated effect of sphingosine against P. aeruginosa was tested only on a sensitive strain (ATCC 27853). Further studies on several other P. aeruginosa strains are necessary to generalize our findings. 

6. Fig. 3: are there any cases where the CFU decreased to 0 after 15 minutes of sphingosine treatment? Without a dot plot, it’s hard to interpret the data. Complete CFU elimination, even in rare cases, is encouraging because untreated P. aeruginosa almost always leads to chronic lung infections. Given the current data, sphingosine is not significantly disrupting lung functions even at 500uM. Would an increase of sphingosine dosage help achieve P. aeruginosa elimination? Please at least add some discussions.

- We thank the referee for this interesting suggestion and now provide the data as dot plot. This presentation indicates the strong effects of sphingosine on bacterial numbers even better than just simply mean ± SD, which is still given. However, in 2 out 5 lungs we achieved complete eradication of P. aeruginosa, which demonstrated the potential of sphingosine. 

- It is possible that we did not achieve complete elimination of P. aeruginosa, because we only inhaled for 10 min. While increasing the dose might also increase the effect on P. aeruginosa, it may also result in alterations of the particle size in the inhalation aerosol. We prefer to inhale a lower dose of sphingosine for longer time in future studies improving the efficacy of sphingosine inhalation.

7. Table 4: it’s quite difficult to distinguish the difference between NaCl and SPH treated groups by looking at the tissue images. The difference in fluorescence intensity, although showed statistical significance, seems small. The authors mentioned that the treatment time might not be long enough for SPH to enter epithelial cells, which could be the case. But I presume the BAL was collected before the lungs are fixed in paraformaldehyde? Have the authors considered using ELISA to quantify the exact concentrations of SPH/ceramide in BAL? Before the lungs are fixed, the authors could also have homogenized some fresh tissue samples and performed a side-by-side SPH/ceramide comparison between BAL and tissue.

- Again, we thank the referee for the recommendation and we have added the analysis of the BAL and provide the data as boxplot. Here, the effectiveness of sphingosine inhalation was clearly demonstrated. 

Reviewer #2: Henning Carstens and co-authors submitted an article on the evaluation of nebulized sphingosine to treat a pig model of perfused lung infected with Pseudomonas aeruginosa. The main result of the study was the possibility of decreasing the bacterial burden in perfused pig lungs by a factor of 6 without observing any side effects.

The study was well conducted and several experiments were performed to evaluate the effect of nebulized sphingosine.

1. My main point is that in general, reductions in bacterial load are expressed in logs, and a 1-log (10-fold) reduction is somewhat considered small. Perhaps this point should be explained in the manuscript?

- It is possible that we did not achieve complete elimination of P. aeruginosa, because we only inhaled for 10 min. While increasing the dose might also increase the effect on P. aeruginosa, it may also also result in alterations of the particle size in the inhalation aerosol. We prefer to inhale a lower dose of sphingosine for longer time in future studies improving the efficacy of sphingosine inhalation.

2. The second point is that the authors showed that sphingosine is more effective than saline in reducing bacterial load, but no comparison was made with an antibiotic commonly used to treat Pseudomonas lung infections, such as ciprofloxacin or nebulized tobramycin.

- We have used P. aeruginosa, since many P. aeruginosa are resistant to many antibiotics including even ciprofloxacin or nebulized tobramycin. We have previously shown (Verhaegh R, Becker KA, Edwards MJ, Gulbins E. Sphingosine kills bacteria by binding to cardiolipin. J Biol Chem 2020;295:7686-96. Seitz AP, Schumacher F, Baker J, Soddemann M, Wilker B, Caldwell CC, Gobble RM, Kamler M, Becker KA, Beck S, Kleuser B, Edwards MJ, Gulbins E. Sphingosine-coating of plastic surfaces prevents ventilator-associated pneumonia. J Mol Med (Berl) 2019;97:1195-1211. Pewzner-Jung Y, Tavakoli Tabazavareh S, Grassmé H, Becker KA, Japtok L, Steinmann J, Joseph T, Lang S, Tuemmler B, Schuchman EH, Lentsch AB, Kleuser B, Edwards MJ, Futerman AH, Gulbins E. Sphingoid long chain bases prevent lung infection by Pseudomonas aeruginosa. EMBO Mol Med 2014;6:1205-1.) that sphingosine kills P. aeruginosa strains regardless of their resistance to classical antibiotics. The present studies served to establish whether sphingosine works in principle and future studies must be designed to improve dosing, efficacy and compare sphingosine with classical antibiotics. 

Minor points

3. Why did you choose a concentration of 500 µM of sphingosine, what was the dose delivered to the lungs?

- In test series we have tested dosages from 125 µM to 500 µM sphingosine, a dosage of 500 µM proved to be sufficient, therefore we started the test series with this dosage.

4. Line 186. I am not sure if the lungs can be inhaled. Perhaps write that the lungs were treated by nebulization of 5 ml of ...

- The text has been changed to “Lungs were treated by a 15-minute nebulization of a 5 mL sphingosine suspension containing 500 µM sphingosine in 0.9% saline or a 

5 ml 0.9 % saline as control”

5. Sphingosine was delivered as a suspension, did you evaluate the efficacy of the nebulization process (total output, particle size change during nebulization…).

- The specified particle size is between 0.4 and 4.4 µm and is therefore well suited for nebulization of our SPH solution. An additional measurement of the inhalation quality was not performed.

6. In Fig. 3, why did you make 2 groups of untreated infected lungs. If all groups received the same, homogeneous initial bacterial inoculum, the CFU counts of the so-called "after infection" groups could be averaged to form a pre-treatment group to compare with the 2 treated groups (saline and sphingosine). 

- As recommended by the first reviewer, we have now implemented a dot-plot with pairwise progression, so it makes sense in our eyes not to change the statistics to that effect now.

7. Again, I think that a group treated with an antibiotic such as tobramycin could have helped evaluate the efficacy of the sphingosine treatment.

- This is a very interesting point, which we will take into account for subsequent studies. Since this experimental setup is quite complex, we have tried to include as few variables as possible.

8. Why sphingosine was assayed only in tissues and not in the BAL? 

- We thank the referee for the recommendation and we have added the analysis of the BAL and provide the data as boxplot. Here, the effectiveness of inhalation was clearly demonstrated. 

9. How long after the nebulization was collected the samples for sphingosine assays.

- The samples were taken 1 hour after nebulization

10. Why were the variabilities in sphingosine, ceramide, and sphingomyelin concentrations higher in the sphingosine group than in the saline group?

 - Statistically they are not higher

11. Lung functional parameters could have been tested on non-infected lungs before and after SPH nebulization to assess its effect on the lungs.

- In our previous work, we tested the effect of sphingosine at different doses on EVLP lungs and could not demonstrate any dose-dependent changes. Therefore, in order to keep the number of animals as small as possible, we did not use this arm.

12. It could have been interesting to test several sphingosine doses and different Pseudomonas aeruginosa strains.

- As written above, the experimental setup was very complex and we tried to include as few variables as possible, but for further series of experiments it is planned to test also other P. aeruginosa strains, other bacteria and changed SPH dosages.

Line 418-419 is not clear to me.

We changed the sentence to: “Our studies demonstrate a significantly increased sphingosine concentration in epithelial cells of infected EVLP measured in fluorescence intensity studies”.

However, more importantly, we now also show quantitative analysis of sphingosine in the bronchoalveolar lavage fluid.

---

## [Editor Report · Decision Letter 1]

5 Jul 2022

Antimicrobial effects of inhaled sphingosine against Pseudomonas aeruginosa in isolated ventilated and perfused pig lungs

PONE-D-22-06001R1

Dear Dr. Henning Carstens,

We’re pleased to inform you that your manuscript has been judged scientifically suitable for publication and will be formally accepted for publication once it meets all outstanding technical requirements.

Kind regards,

Abdelwahab Omri, Pharm B, Ph.D, Laurentian University, Canada

Academic Editor

PLOS ONE

---

## [Editor Report · Acceptance letter]

12 Jul 2022

PONE-D-22-06001R1 

Antimicrobial effects of inhaled sphingosine against *Pseudomonas aeruginosa* in isolated ventilated and perfused pig lungs 

Dear Dr. Carstens:

I'm pleased to inform you that your manuscript has been deemed suitable for publication in PLOS ONE. Congratulations! Your manuscript is now with our production department. 

Kind regards, 

on behalf of

Dr. Abdelwahab Omri 

Academic Editor

PLOS ONE